# The lived experience of people affected by cancer: A global cross-sectional survey protocol

Julie Cayrol[1,2,3]*, Claire E. Wakefield[4,5], André Ilbawi[3], Mark Donoghoe[5,6], Ruth Hoffman[7], Moses Echodu[8], Clarissa Schilstra[4,5], Roberta Ortiz[3], Lori Wiener[9]

1 The Royal Children's Hospital and Murdoch Children's Research Institute, Melbourne, Australia, 2 Department of Paediatrics, Faculty of Medicine, The University of Melbourne, Melbourne, Australia, 3 World Health Organization, Department of Non-Communicable Diseases, Geneva, Switzerland, 4 School of Clinical Medicine, UNSW Medicine & Health, Randwick Clinical Campus, Discipline of Paediatrics, UNSW Sydney, NSW Sydney, Australia, 5 Behavioural Sciences Unit, Kids Cancer Centre, Sydney Children's Hospital, Randwick, Australia, 6 Stats Central, Mark Wainwright Analytical Centre, UNSW Sydney, NSW Sydney, Australia, 7 American Childhood Cancer Organization, Beltsville, Maryland, United States of America, 8 Uganda Child Cancer Foundation, Kampala, Uganda, 9 National Cancer Institute, Center for Cancer Research, National Institute of Health, Bethesda, Maryland, United States of America

* Julie.Cayrol@mcri.edu.au

**Data Availability Statement:** No datasets were generated or analysed during the current study. All relevant data from this study will be made available upon study completion.

## Abstract

A diagnosis of cancer impacts the person's physical and mental health and the psychosocial and financial health of their caregivers. While data on the experience of living with cancer is available, there is a dearth of data from persons in low- and middle-income countries (LMICs). The perspectives of other impacted individuals also remain understudied (e.g., bereaved family members), as well as the impact on survivors and their families over time. The objective of this study is to describe the psychosocial and financial impact of cancer on people diagnosed with cancer as a child, adolescent or adult, their families/caregivers, and the family members of those who have died from cancer, in high-income countries (HICs) and LMICs. This study is an observational, descriptive, quantitative study. Data will be collected anonymously via a digital online cross-sectional survey distributed globally by the World Health Organization (WHO) via the LimeSurvey software. Participants will include (a) adults aged 18+ who have been diagnosed with cancer at any age, who are currently undergoing cancer treatment or who have completed cancer treatment; (b) adult family members of individuals of any age with a cancer diagnosis, who are currently undergoing cancer treatment or who have completed cancer treatment; and (c) bereaved family members. Participants will be anonymously recruited via convenience and snowball sampling through networks of organisations related to cancer. Survey results will be analysed quantitatively per respondent group, per time from diagnosis, per disease and country. Results will be disseminated in peer-reviewed journals and at scientific conferences; a summary of results will be available on the WHO website. This study will suggest public health interventions and policy responses to support people affected by cancer and may also lead to subsequent research focusing on the needs of people affected by cancer.

**Funding:** The authors received no specific funding for this work. WHO will receive financial support from the American Childhood Cancer Organization (ACCO) for this research and other projects. Ruth Hoffman (ACCO) collaborated in the study design and preparation of the manuscript as a mother of a childhood cancer survivor but will not have a role in the data collection, analysis, preparation of future manuscripts or decision to publish.

**Competing interests:** The authors have declared that no competing interests exist.

# Introduction

## Background

Cancer is a leading cause of illness and mortality. In 2020, nearly 20 million people developed cancer, and nearly 10 million deaths were caused by cancer [1]. Regardless of setting or context, a diagnosis of cancer marks the start of a long and complex journey for the person living with cancer and their family. This journey impacts not only the person's physical health but also their mental health, as well as the psychosocial [2] and financial health of the family unit [3, 4].

Documenting the lived experience of people affected by cancer is the first step to identifying how services can be organised to meet their needs. This is an essential prerequisite to implementing integrated people-centred care which aims to put people and their communities at the centre of health systems [5]. The 'lived experience of people affected by cancer', or 'lived cancer experience', is defined here as the experience of living or having lived with cancer, including, but not limited to the experience of receiving treatment for cancer and/or supporting someone through cancer and beyond, in the long-term after treatment has finished.

The psychosocial dimensions of the impact of cancer can be organised into the following categories, which are themselves interlinked: psychological, social, and financial. These are described below.

**Psychological impact of cancer.** Mental health disorders such as adjustment, anxiety and depressive disorders, secondary to cancer treatment are commonly reported in review articles [2, 6, 7], A meta-analysis of 94 interview-based studies reported that a combination of mood disorders could occur in up to 30–40% of adults, in hospital oncological and palliative care settings, although the vast majority of these data are collected in HICs [7]. The psychosocial morbidity associated with cancer increases the burden of disease and is associated with a reduction in quality of life, poor adherence to treatment and possibly, decreased chances of survival [6, 7].

A systematic review of the prevalence of depression and anxiety among people at least 2 years after a cancer diagnosis, found that anxiety was more prevalent among cancer survivors (17.9%; 95% CI 12.8–23.6) compared to healthy controls (13.9%; 95% CI 9.8–18.5; with a RR of 1.27 (95% CI 1.08–1.50; p = 0.0039). This study also found that anxiety was common among spouses of cancer survivors, with a prevalence of anxiety in survivors of 28.0% (95% CI 22.3–33.9) and 40.1% among spouses (95% CI 25.4–55.9) [8]. Most participants from this study were from HICs.

Multiple studies have reported symptoms of depression, anxiety, post-traumatic stress disorder (PTSD), behavioural problems, and drug and alcohol misuse in survivors of childhood and adolescent cancer at higher rates than their peers [9–12]. In parents of childhood cancer survivors, two systematic reviews report that worry and disease-related thoughts, anxiety, PTSD and depression also occur more frequently than in controls, though again, data is predominately from HICs [13, 14]. Finally, a systematic review of studies in HICs comparing psychosocial functioning in siblings of childhood cancer survivors with their peers found significantly more symptoms of PTSD and poorer academic performance as well as greater likelihood to engage in risky health behaviours in adulthood [15], especially for those who report less social support [15].

The limited data from LMICs shows impaired health-related quality of life one year after diagnosis in 8 countries in Southeast Asia [16] and rates of depression and anxiety of up to 21% and 18% respectively in adults with cancer [17]. However, there remains a substantial lack of research in many LMICs in comparison to HICs, on the mental health outcomes of long-term survivors and of certain groups of people affected by cancer, such as children, and their

family members. Even in HICs, certain vulnerable populations of cancer survivors and their families, such as rural and disadvantaged socioeconomic populations, experience high rates of anxiety, depression and distress [18] and an elevated and prolonged need for mental healthcare [19], yet their needs are yet to be explored on a global level.

Grief is an additional need and an anticipated response following a cancer diagnosis or death of a loved one. In certain situations, when the grieving process becomes prolonged, it may lead to prolonged grief disorder, which can itself contribute to significant challenges in psychosocial functioning [20–22]. Although bereaved parents of children with cancer demonstrate resilience, protracted grief reactions such as prolonged grief disorder [23] are associated with poor quality of life and psychological and physical health outcomes, including more mortality [22, 24–27]. A systematic review of studies from HICs examining the psychosocial outcomes of cancer-bereaved children and adolescents shows high levels of adjustment, though some studies did show frequent unresolved grief and a higher risk of self-harm in bereaved adolescents [28]. Most of the data examines response in the medium-term, but data in the long-term, particularly 10 or more years after death, is lacking.

Importantly, a cancer diagnosis and treatment can also result in positive psychological outcomes. Survivors and their families have, for example, reported increased resilience, post-traumatic growth, positive health behaviour changes, and increased benefit-finding after diagnosis and treatment [28–31].

**Social outcomes.**  Cancer has an important impact on the social lives of people of all ages. Systematic reviews have reported that adults with cancer may experience social difficulties, such as loneliness and social isolation [32, 33], challenges in their marital/partner relationships [34, 35] and sexual dysfunction [36]. They may benefit from targeted psychosocial interventions to improve social function and quality of life, as demonstrated in a meta-analysis of 22 randomized controlled trials [37]. A systematic review examining the social and economic impact of cervical cancer on women and children in LMIC demonstrated mixed results in terms of social support and independence, with most qualitative studies reporting social isolation, strained relationships with their partners, and a negative impact on independence, but also in some quantitative studies, high levels of social support [38].

When it comes to adolescent and young adults (AYAs) with cancer, a systematic review of studies mostly in HICs shows that AYAs aged 15–39 can experience more problems in social functioning and have poorer social quality of life than their peers without cancer [39]. This includes problems with employment, with the maintenance and development of peer and family relationships, intimate and marital/partner relationships, and needing peer support, even years after diagnosis [39, 40]. A systematic review examining the determinants of social functioning among AYAs with cancer shows that adequate social support is associated with increased social functioning in this population [41]. Most of the data on child and adolescent cancer arise from HICs and do not extend into the long-term.

**Financial toxicity.**  Financial toxicity and financial hardship are used interchangeably to describe the distress caused by the economic burden of cancer care [42, 43]. Nearly half of all people living with cancer experience a financial impact, which is more pronounced in people who lack health insurance, have a lower income, are unemployed, come from a minority ethnicity, or are younger at diagnosis [3, 44–46]. Financial toxicity is closely linked with financial distress and can lead to the decision to forego care [45]. Two systematic reviews [47, 48] studied financial toxicity in HICs with a publicly funded healthcare, one of them including qualitative studies only [48]. Direct material hardship and a reduction in financial resources is related to out-of-pocket expenses for medical care and indirect costs such as those associated with travel or housing during treatment, loss of income and assets and medical debts, with the majority of studies focusing on patients rather than caregivers [47, 48]. Survivors also face

significant costs of long-term access to care for chronic conditions secondary to cancer treatment, as demonstrated in a survey of 1656 cancer survivors in the USA [49].

While there is limited data from studies conducted in LMICs, available data suggests a very significant financial impact, with many families reporting a need to ask for money from charity, borrow money, or sell possessions to access healthcare [50, 51]. A longitudinal study of 9513 adults with cancer in Southeast Asia found that economic hardship was reported by up to a third of families and that 48% of adults had experienced financial catastrophe 12 months after diagnosis (defined as out-of-pocket costs equal to or exceeding 30% of annual household income). The risk was increased in people of low income groups and in those without insurance or whose insurance does not cover the costs of treating cancer [52, 53]. In LMICs, the economic impact of cervical cancer on women affected has been reported as a decline in the standard of living, a need to rely on family members as well as loss of hours of employment. However, the indirect costs of cancer and the broader effects on family are yet to be better described [38].

Productivity loss is another factor contributing to economic hardship, which extends into survivorship [49]. Productivity loss results from lost employment, prolonged leaves of absence due to medical appointments and hospitalizations, and an inability to participate in work-related activities or to re-enter the workforce after treatment, contributing to financial toxicity [4, 45, 46, 48]. A study conducted in the BRICS countries (Brazil, Russia, India, China, and South Africa) estimated the lost productivity due to premature cancer mortality in 2012 to reach 46.3 billion US dollars, with the largest total productivity loss seen in China [54]. This study was completed over a decade ago, and given the increased global cancer prevalence, ageing and growth of the population [55], this loss is likely to be significantly higher today.

For survivors of childhood and adolescent cancer, studies from HICs suggest that they are less likely to graduate from university, be employed, or live independently compared with peers or siblings without a cancer history [56–60]. They have also reported to have lower wages and require social security for income supplementation [61]. Australian data suggests that the lifetime productivity loss of AYAs aged 15–25 years diagnosed with cancer in 2016 will be $508.4 million AUD. These costs result from productivity losses associated with absenteeism from school, lower workforce participation, premature mortality, informal carer costs and administrative costs [62]. Data from the USA suggests costs from productivity losses of AYAs aged 15–39 years diagnosed with cancer total $18.3 billion USD [63].

Many families of a child with cancer must manage the direct and indirect costs of treatment whilst simultaneously needing to reduce their working hours to care for their child. Effects on income losses can be ongoing, for years after treatment ends. Families of low socioeconomic background, mothers who were on maternity leave at the time of their child's diagnosis, and single mothers are at greatest risk of employment loss [64–66]. Adult siblings of childhood cancer survivors in the United States are also more likely to suffer diverse forms of financial hardship such as difficulty paying bills or foregoing medical care, than the general population [67]. Importantly, in a systematic review of the socio-economic impact of childhood cancer on parents, only 6 of the 29 studies were from LMICs in Asia and Africa, further highlighting the lack of data available in LMICs [66].

Finally, financial hardship due to cancer increases the risk of psychological consequences, such as social isolation and elevated stress and worry, [45, 47, 48, 64] across the lifespan and is associated with lower health-related quality of life [68]. Assessing risk for financial hardship from the time of diagnosis throughout the cancer trajectory is an evidence-based standard for psychosocial care of children and adolescents with cancer [69].

## Rationale

Overall, the effects of cancer are broad. However, data on the lived cancer experience from LMICs, and data comparing psychosocial effects to those experienced by people in HICs, particularly in the long-term after cancer treatment, is lacking. This is particularly important given there are likely fewer psychosocial and financial supports available for people with cancer in LMICs.

Understanding the immediate and long-term lived cancer experience of people on a global level, particularly in areas of unmet need, would allow identification of key targets for health systems and policy intervention. Strengths and novelties of this study include its global nature, its availability in different languages and the fact that it targets individuals and family members during and after cancer treatment.

## Material and methods

### Aims

The aims of the study are to understand and describe the immediate and long-term psychosocial and financial impact of cancer on people diagnosed with cancer as children and as adults, their families/caregivers and the family members of those who have died from cancer in HICs and LMICs.

### Design

This study is an observational, descriptive, quantitative, cross-sectional study with some open-ended questions, distributed by the World Health Organization (WHO) Headquarters amongst all member countries and regions [70].

### Governance

The study was designed by a 9-member Steering Committee representing people with lived experience of cancer (survivors and parents), professional disciplines (oncology, psychology, social work, statistics, research, clinical trials), organisations (WHO, community organisations, universities, hospitals) and regions (US, Europe, Africa, Australia). Throughout the course of the project, the Committee met monthly via videoconference, finalising study aims, developing the study design, drafting the study protocol and survey questionnaire, and contributing to project management. The Committee is also responsible for collecting stakeholders' and initial participants' feedback and finalising the survey questionnaire. The Committee will advise on policy impacts of the project and ensure that translation and sustainability are at the forefront of planning. The Committee will disseminate newsletters and annual reports to all stakeholders. Following project launch, the Committee will invite additional members as needed, including, for example, underrepresented groups of people with lived experience (e.g., fathers, culturally diverse people, First Nations people) through direct contact with cancer organisations and institutions prior to re-launch, to ensure that the project is culturally safe and inclusive. We will continue to invite additional professional experts as needed (Fig 1).

The Steering Committee is supported by a larger Advisory Panel, comprising up to 50 stakeholders who provided feedback on survey questionnaire content and length, participate in study webinars, and promote the study to their networks, amongst other activities. Advisory Panel members include people with lived experience of cancer, representatives from relevant cancer community organisations around the world, WHO staff in regional offices, members of the Global Childhood Cancer Initiative [71], and other individuals as relevant. The study is

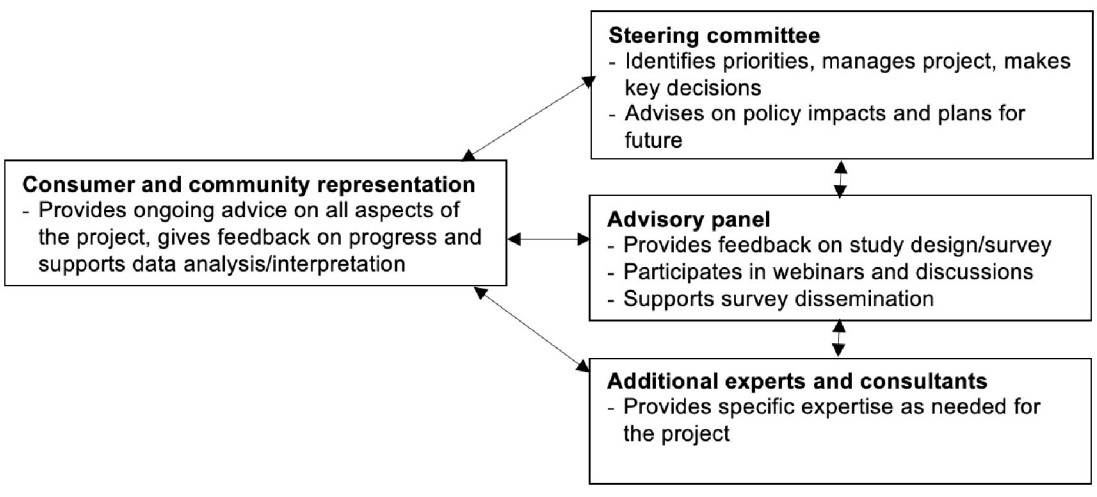

**Fig 1. Project governance.**

supported by additional experts and consultants (e.g., graphic design, communications, translation services) as needed.

## Measures

The survey questionnaire was developed by the Steering Committee utilising the WHO Quality of Life framework, with additional questions included from a series of validated tools. The research team also purposely developed additional items where validated items were not available. Some open-ended questions were added to elicit valuable insights and to capture the voices of survivors and family members, particularly on issues the survey questionnaire may have missed. The content of the survey questionnaire was shared with the Advisory Panel engaged in this project to ensure that important medical, psychosocial, and financial aspects were included to ensure cultural sensitivity. A summary of the structure and content of the survey questionnaire, with a list of the different topics addressed, the items included, the measure and response options, and the participants to whom each item is directed is provided (Table 1; comprehensive S1 Table). The comprehensive table is available as supplementary material.

## Ethics and safety considerations

Ethics approval was obtained through the WHO Ethics Review Committee (ERC). This study was deemed exempt from formal review by the Committee, given its low-risk nature. The survey questions are designed for general use across HICs and LMICs, so no ethical approval will be sought from individual participating countries.

The Steering Committee considered the priorities and needs of populations in low resource settings, as outlined in the Council for International Organizations of Medical Sciences (CIOMS) Ethical guidelines on Research conducted in low-resource settings [72], the principles of which are met in this study. Electronic consent is included within the online survey.

## Participants and recruitment

Four groups of participants will be recruited:

**Table 1. Summary of items in survey questionnaire.** Aims, measures and target participants.

| Aim | Measure and response option | All People affected by cancer | Survivors | Family members of people affected by cancer | Bereaved family members |
|---|---|---|---|---|---|
| **Introductory questions** | Forced choice options. Items purposely developed. | X | X | X | X |
| **Demographics** | Mixture of forced choice options with options to add free text (e.g., 'other'), depending on the question. Items either purposely developed or based on Childhood Cancer Survivor Study or the FOCUS study. | X | X | **About self** | **About self** |
| **Clinical information about person with cancer** | Mixture of forced choice options with options to add free text (e.g., 'other'). Items based on the FOCUS study. | X | X | **About their family member (excluding type of treatments received)** | **About their family member (excluding types of treatment received)** |
| **Survivorship care experiences** | Items based on the FOCUS study | X | X | - | - |
| **Fertility experiences of person with cancer** | Mixture of forced choice options with options to add free text (e.g., 'other'). Items purposely developed. | X | | - | - |
| **Perceived impact of cancer** | Items based on the FOCUS study | X | X | - | - |
| **Impact on marriage/ relationship (if applicable)** | Purposely developed | X | X | X | X |
| **Education/work problems** | Adapted from Long Term Follow Up study questionnaire and purposely developed | - | X | X | X |
| **Caregiver burden** | Family Appraisal of Caregiving Questionnaire for Palliative Care (FACQ-PC) by Cooper et al. 2006 [75], and an additional item purposely developed | | | **Self** | **Self** |
| **Prolonged Grief** | Prolonged Grief Scale by Prigerson et al. 2021 [76] | - | | - | **Self** |
| **Impact of cancer on siblings** | Sibling Cancer Needs Inventory (Patterson et al. 2014) [77] and purpose-designed questions (11 items) | - | | **Siblings only** | - |
| **Health-related quality of life** | PROMIS Global | X | | **Self** | **Self** |
| **Health behaviours** | Purposely developed | X | X | **Self** | **Self** |
| **Emotional problems** | Purposely developed | X | X | **Self** | **Self** |
| **Mental health** | Purposely developed | X | X | **Self** | **Self** |
| | Items based on the FOCUS study | X | X | **Self** | **Self** |
| | PROMIS Anxiety short form | X | X | **Self** | **Self** |
| | PROMIS Depression short form | X | X | **Self** | **Self** |
| **Financial well-being** | Purposely developed | X | X | **Self** | **Self** |
| | COST measure developed by De Souza et al 2017 [78] | - | X | **Self** | **Self** |
| | COST Measure, developed by De Souza et al 2017 [78] | X | | **Self** | **Self** |
| **Support service use** | Based on a National Cancer Institute Survey | X | X | **Self** | **Self** |
| **End of life care experience** | Purposely developed | - | - | - | **Self and family member** |
| **Post-traumatic growth** | Purposely developed | X | | **Self** | **Self** |
| **Missed topics** | Purposely developed | X | | **Self** | **Self** |

i). Adults currently living with cancer (with a cancer diagnosis or undergoing cancer treatment)

ii). Survivors of childhood, adolescent, or adult cancer, who have completed cancer treatment

iii). Family members or caregivers of people with cancer or long-term survivors

iv). Bereaved family members or caregivers who have lost a family member to cancer.

The inclusion criteria is defined as an individual who:

- Is an adult older than or equal to 18 years of age and

- Has had a cancer diagnosis (in childhood or adulthood): this includes adults who are currently undergoing treatment, and who have finished treatment or are not currently receiving treatment (including survivors of childhood, adolescent, young adult or adult cancer).

Or as an individual who:

- Is 18 years of age or older and

- Is a family member (spouse, child, sibling, or other primary caregiver) of a person ever diagnosed with cancer or

- A family member (spouse, child, sibling, or other primary caregiver) of a person who has died from cancer

The exclusion criteria are defined as:

- Children under 18 years of age. These children will be represented by their parent or guardian.

- Siblings under 18 years of age of a person with a cancer diagnosis.

Non-English speaking people will be invited to participate. The survey questionnaire will be translated in the five other United Nations languages (Spanish, French, Chinese, Russian, Arabic) through the WHO Headquarters Interpretation services. Other languages may also be represented depending on availability of translators. After translation, the survey questionnaire will be reviewed by native speakers amongst the study team, where possible, to ensure that translations are adapted to culture, context, and use of language. The survey questionnaire will be reviewed within WHO by a staff member with Health Literacy expertise to ensure the content is adequate for health resource-constrained countries.

Participants will be anonymously recruited via convenience and snowball sampling through networks of organisations related to cancer. To date, the survey has been shared within the WHO and with partner networks of international cancer organisations via a webinar, direct meetings, as well as through direct emails to organisation leaders. Cancer organisations that were chosen to disseminate the survey focus on childhood and on adult cancer (including specific cancers such as women's cancer), with a broad geographical representation. These include professional partner organisations, research institutes, as well as foundations and advocacy groups (S2 Table). Organisations will be permitted to advertise the survey on social media a maximum of three times. Physicians and other healthcare providers have not directly been approached to identify participants.

## Follow-up

At the end of the survey, participants will be able to opt-in to a possible future in-depth qualitative interview by providing their contact details (name, email address, phone number) to an email address that will be provided to them. This will allow the research team to explore topics identified from the quantitative data in more depth. Participants will also have the option of providing their personal details to receive an easy to understand summary of results (see 'Dissemination of results' below).

If participants do not provide their contact details through that route, no follow-up will occur with participants. Participants will not be contacted individually with results from this survey, however an easy-to-read summary of the results of the survey will be published on the

WHO website. While this survey may not result in direct benefit to individuals enrolled, participants may appreciate the opportunity to share their experience regarding how cancer has impacted their health and their current life. We also anticipate this study will have broader public health benefits.

## Timeline and status

A pilot version of the survey was launched in October 2022, after obtaining Ethics approval and appropriate stakeholder feedback on the content and format of the questions. The pilot survey was launched in English, French and Spanish. The Steering Committee used pilot findings and stakeholder and participant feedback on the pilot survey to develop the final version of the survey. The final survey will be available in other languages too as mentioned above. To increase representation from people in LMIC and men, who were under-represented in the pilot sample, additional direct meetings and communication were undertaken with stakeholders from the Advisory Committee to improve dissemination and participant recruitment. We anticipate relaunching the survey in October 2023, completing data collection and beginning data analysis and synthesis in the first quarter of 2024.

## Data management and statistical analysis

### Data management and confidentiality

All data will be collected, securely stored, and managed confidentially and locally within WHO, via the LimeSurvey software [73]. Identifiable data will only be collected and stored from participants who request follow-up by sending an email to the study group. Personal data will be linked to a newly created personal ID number in a document stored on a secure WHO server with access permitted only by the study team. Follow-up with participants who have provided their details will be done by members of the study team only. All personal data will be destroyed 5 years after publication of results.

Study data will only be accessible to the research team and will not be shared with any other external organisation or individual during the data collection period. All data will be digital, and no hard copies will be stored. All data will be stored for a total of five years from publication, after which it will be destroyed.

### Data analysis

The survey results will be analysed quantitatively per respondent group (people affected by cancer, their families and carers), per time from diagnosis, as well as by disease and age group (e.g.: paediatric neuro-oncology, paediatric leukaemia) and by country.

Quantitative variables will be summarised using means and standard deviations or medians and interquartile ranges, as appropriate. Categorical variables will be summarised using frequencies and percentages. Summary statistics will be presented along with a 95% confidence interval, to reflect the uncertainty in their estimation.

The anticipated total sample size is 2000 respondents. Within a particular subgroup, a sample size of 300 independent responses would provide a 95% confidence interval for the mean of a quantitative variable with a margin of error of ± 0.11 standard deviations. For the proportion associated with a categorical response, the margin of error would be no larger than ± 6%. Where it is possible to identify non-independent groups of responses (e.g., participants recruited via snowball sampling), standard errors will account for the clustering.

It should be noted that the sampling approach will be subject to both selection and non-response biases, as some of the target population will not be aware of the questionnaire, and

others will choose not to respond. In order to assist with interpretation, where possible, the basic demographic characteristics of respondents will be contrasted to those of the target population, and the characteristics of early versus late respondents will be compared [74].

Additional limitations to a cross-sectional design include the wide range of participants included in this study (age, different diseases, time since diagnosis, patients, caregivers and bereaved). Finally, the aim of this study is to describe the current situation of people with a lived cancer experience. This study will not attempt to derive causal relationships between a cancer diagnosis and participants' current or past lived experience.

A detailed analysis plan will be prespecified prior to undertaking analyses to address more complex research questions, for example, to define regression models that explore the interrelationships between different survey variables.

## Outcomes and dissemination

### Study outcomes

Outcomes of this study will include a description of different aspects of the lived cancer experience, in HICs and LMICs (Table 2).

### Implications for future research

Results from this study will allow WHO and the research team to provide a description of the lived experience of people affected by cancer globally, particularly focusing on their psychosocial and financial needs. This has the potential to contribute to the development of legislative and policy interventions for professionals and governments designed to support people living with cancer and their families, led by WHO. This study may also lead to subsequent research questions focusing on the needs of people affected by cancer and their family members and caregivers, in different nations.

### Dissemination of results and publication policy

A webinar will be conducted before the publication of the results. The objectives of the webinar will be to disseminate results to professional organisations, regional WHO offices, cancer organisations, grassroot organisations and to the Advisory Panel, to raise awareness and to

**Table 2. Study outcomes.**

| A description of: |
| --- |
| 1. Immediate and long-term financial impact of cancer on people affected by cancer and their family members. |
| 2. Physical health and quality of life of people affected cancer and their family members. |
| 3. Emotional well-being, mental health, and psychological supports of people affected by cancer and their family members. |
| 4. Social well-being (relationships, work, and education) of people affected by cancer and their family members. |
| 5. The nature of information received by people affected by cancer and their family members, and the degree of satisfaction in relation to the information received. |
| 6. Survivors' and their family members' perception of quality of cancer care during and after treatment. |
| 7. Survivors' opinions and experience accessing follow-up and survivorship multidisciplinary care. |
| 8. Siblings' experience of cancer. |
| 9. Caregiving experience of family members/caregivers and the impact of caring for someone with cancer on family members/caregivers. |
| 10. Bereaved family members' experience of cancer. |

commence translating the results into practical recommendations. Future implementation studies may allow us to monitor the implementation of these recommendations.

Results will also be disseminated through peer-reviewed publications and through presentations at appropriate conferences and seminars. A synthesis of the findings will allow WHO and the research team to create a WHO practice brief on the lived experience of people affected by cancer. The brief will provide practical recommendations for Member States and suggest possible international collaborations to meet those needs; setting the agenda for person-centred cancer care. This practice brief and recommendations will be available on the WHO website. An easy to understand summary of results will be shared with participants who wish to receive this via email (see 'Follow-up' section).

## Supporting information

**S1 Table. Items and measures in survey questionnaire.** Complete list of items, measures and response options in survey questionnaire.
(DOCX)

**S2 Table. Stakeholders supporting study dissemination.** Stakeholders per geographical region and focus population.
(DOCX)

## Acknowledgments

We would like to acknowledge Mark Donoghoe for this contribution to data analysis; the stakeholders and cancer survivors who have provided feedback on this survey; and the American Childhood Cancer Organization. This research was supported in part by the National Cancer Institute Intramural Research Program of the NIH.

## Author Contributions

**Conceptualization:** Julie Cayrol, Claire E. Wakefield, André Ilbawi, Ruth Hoffman, Clarissa Schilstra, Roberta Ortiz, Lori Wiener.

**Funding acquisition:** Ruth Hoffman.

**Methodology:** Julie Cayrol, Claire E. Wakefield, Mark Donoghoe, Clarissa Schilstra, Lori Wiener.

**Project administration:** Julie Cayrol.

**Resources:** Julie Cayrol.

**Supervision:** Claire E. Wakefield, Lori Wiener.

**Writing – original draft:** Julie Cayrol, Lori Wiener.

**Writing – review & editing:** Julie Cayrol, Claire E. Wakefield, André Ilbawi, Mark Donoghoe, Ruth Hoffman, Moses Echodu, Clarissa Schilstra, Roberta Ortiz, Lori Wiener.

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
