## [Decision Letter · Decision Letter 0]

24 Aug 2023

PONE-D-23-17936The lived experience of people affected by cancer: A global cross-sectional survey protocolPLOS ONE

Dear Dr. Cayrol,

Thank you for submitting your manuscript to PLOS ONE. After careful consideration, we feel that it has merit but does not fully meet PLOS ONE’s publication criteria as it currently stands. Therefore, we invite you to submit a revised version of the manuscript that addresses the points raised during the review process. Please revise the manuscript according to the reviewers’ comments mainly in the method section others.

We look forward to receiving your revised manuscript.

Kind regards,

Sefonias Getachew, MPH, PhD

Academic Editor

PLOS ONE

Reviewers' comments:

Reviewer's Responses to Questions

**Comments to the Author**

1. Does the manuscript provide a valid rationale for the proposed study, with clearly identified and justified research questions?

Reviewer #1: Partly

Reviewer #2: Yes

2. Is the protocol technically sound and planned in a manner that will lead to a meaningful outcome and allow testing the stated hypotheses?

Reviewer #1: Yes

Reviewer #2: Yes

3. Is the methodology feasible and described in sufficient detail to allow the work to be replicable?

Reviewer #1: Yes

Reviewer #2: Yes

4. Have the authors described where all data underlying the findings will be made available when the study is complete?

Reviewer #1: Yes

Reviewer #2: No

5. Is the manuscript presented in an intelligible fashion and written in standard English?

Reviewer #1: Yes

Reviewer #2: Yes

6. Review Comments to the Author

You may also provide optional suggestions and comments to authors that they might find helpful in planning their study.

Reviewer #1: This protocol paper reported a proposed internationally collaborative study which aimed to investigate the lived experience of different groups of people affected by cancer. This study is of importance and significance in understanding the challenges and needs of these vulnerable groups of people, and accordingly, provide evidence for the development of intervention program and health policy.

This protocol paper is well structured, and the study design is aligned with its objectives. Participants, in/exclusion criteria and recruitment were clearly and properly reported.

Below are my comments for the Editor and Authors to consider:

1. It is unclear if the proposed study will be an epidemiological cross-sectional survey, as mentioned in the title and background section, or it is a mixed method study consisting of cross-sectional survey and a qualitative study, as mentioned in the Design (line 265) and Follow-up (since line 364). Please make this clear, and make it consistent throughout the manuscript.

2. The Abstract had sufficient background information, but lack basic information of the proposed study, such as aims, study design, participants and recruitment methods, data collection and analysis and outcome distribution. Please add.

3. Study outcomes (primary and secondary) were not presented in this protocol. Please add.

4. Line 259-63 of page 13, the two secondary objectives were not technically objectives, but more like implications for future research.

5. Line 409-415, contents regarding the sample size estimation need to be rigorously revised after consulting a statistician. It mentioned that “would provide a 95% confidence interval for the mean of a quantitative variable with a margin of error of ± 0.11 standard deviations”. Please note that quantitative variables with different variance require varied sample size to achieve a stable estimate of 95% CI (the more various, the bigger sample size need to be). Therefore, the authors need to specify, as well as to justify, their estimates on the means, SD, error level and anticipated response rate for the sample size estimation. For a global survey, I highly doubt the sample size of 2000 is justifiable.

6. Line 181-2 of page 9: “Limited data from LMIC has found that the financial impact of cancer is as significant as in HIC”. I doubt if this is the fact, as it’s widely known that people from LMICs generally have poorer access to healthcare services due to financial affordability of costs for cancer treatment.

7. Line 250-1 of page 12, the description of the study aim as “The aim of this study is to better understand and document the lived experience of people affected by cancer” is quite vague and lack of focus.

8. Some expressions of the manuscript are ambiguous and need to be more precise, below are some examples:

1) Line 198-9 of page 10: “this loss is likely to be significantly higher today.” it’s good to give your reasons for the assumption, e.g., increased cancer prevalence and aging of the population?

2) line 206 of page 10: “Australian data suggests that for adolescent and young adults diagnosed with…”. And you also mentioned at line 211 of page 11:”… adolescents and young adults diagnosed with cancer total…”. Globally, there are no commonly accepted age ranges for “adolescents and young adults”, so it’s important for you to give the specific age ranges of participants in these two cited studies.

3) Line 220-1 of page 11: “…, and single mothers are at greatest risk (56–58).” This sentence is incomplete, at the greatest risk of what? financial hardship? Income loss?

4) What are the differences/similarities of financial toxicity/hardship?

5) Line256 of page 12:” died from cancer in a diversity of settings (HIC and LMIC).” By “settings”, do you mean in countries with different levels of development?

6) Line 265-7 of page 13: “This study is an observational, descriptive, mixed method cross-sectional study, distributed globally by the World Health Organization (WHO) Headquarters. No particular country or region is specifically targeted for inclusion.” Do you mean “This study is an … survey, which will be distributed among all member countries or regions of The World Health Organization (WHO), with no particular exclusion of any country or region.”

7) Line 314-5 “the questions are not targeting specific countries. For this reason, ethical approval from other countries was not sought.” A more rigorous description could be “survey questions are designed for general use across HICs and LMICs, so no ethical approval will be sought from individual participating countries.”

9. The introduction section focused on the findings of previous studies on three major impacts of cancer on different groups of populations. Nevertheless, this section failed to adequately explain why the proposed study is needed.

It is necessary to systematically review existing literature on relevant topics, e.g. what existing surveys/studies have investigated, the methodologies they used, main findings (this was included in the current manuscript). Additionally, it is important to point out the gaps in current literature and how your proposed study will meet or help to narrow these gaps. It’s also crucial in this section to clarify the strengths of your proposed study compared to previous studies. Otherwise, it would be questionable in the necessity or significance of the proposed study. For example, if there is no adequate novelty of your proposed study, then to conduct a systematic review may be enough to get a comprehensive knowledge of this topic.

10. Line 277-8 of page 13: “finalising the survey, and interpreting the data after data collection.” This is confusing. Do you mean you have collected any survey data?

11. Authors misused “survey” as “survey questionnaire” in multiple places, e.g., line 299 of page 14 “The survey was developed by the Steering …”, line 304 “… the survey may have missed.” The “survey” in these sentences should be “survey questionnaire”. Please check and correct thoroughly.

12. Line 307 “Table 1 summarizes the structure and content of the survey.” Here, it’s essential to report the main information of Table 1 in texts to facilitate understanding of readers. Also, it probably will be difficult to present all the information of Table 1 in the main text when it comes to publication. I suggest keeping the outline and main contents of Table 1 in the main text, while keeping the completed version as a supplement. The title of Table 1 contains repeated information, consider to revise it.

13. For such a global survey, it’s crucial to take into consideration of the priorities and special needs of people from health resource constrained countries, from non-English speaking countries, or with special cultural beliefs of cancer (e.g., indigenous people). This was touched in line 317-20 briefly, but it’s not clear how these important aspects will be addressed in the study design. Please add relevant information. Similarly, more information is needed how the study will be implemented in country with different languages, cultures and health systems? Will limited adaptation of the survey questions allowed? Will translation in local language available?

14. Some people in under-developed countries may have no or very limited access to medical treatment. It can be important to know if they are able to receive treatment, if so, is this treatment adequate (i.e., to have the necessary treatments that they need); if not, what are the key barriers.

15. Line 366-7 “… by providing their contact details (name, email address)”. Authors may consider including phone number as a contact method, as this is more commonly and frequently used than email. By including name and email only, it may be at a risk of omitting certain groups of people who don’t use/check email regularly, like those from low socioeconomic status.

16. Other minor changes

1) Line 268, to remove “(Figure 1)” from the subtitle, as you will refer to it in the main text.

2) All HIC and LMIC to be HICs and LMICs

3) At line 99-102 of page 6:” A systematic review of the prevalence of depression and anxiety among long term cancer survivors found that anxiety was more prevalent among cancer survivors compared to healthy controls, and was common among spouses of cancer survivors”. Here you may specific the definition of “long term cancer survivors”, as well as the prevalence of the two study groups.

4) Line 423 “Additional limitations to a cross-sectional design include difficulty deriving causal relationships between a cancer diagnosis and participants’ current or past lived experience”. If this is going to describe the current situation, rather than to perform causal inference, then the mentioned limitation would not be an issue.

Reviewer #2: 1 The rationale and research questions are valid

2/3 The protocol is technically and methodically sound, it can be optimized a bit

4 They have a segment about data handling that currently lacks a statement about making the data available. It is recommended to add

5 Yes, it was interesting and easy enough to read and understand. the background can be condensed a bit

7. PLOS authors have the option to publish the peer review history of their article (what does this mean?). If published, this will include your full peer review and any attached files.

Reviewer #1: **Yes: **Shurong Lu

Reviewer #2: **Yes: **Eric S Kroeber

---

## [Author Response · Author response to Decision Letter 0]

4 Oct 2023

Dear editors and reviewers, 

We thank the reviewers for their feedback on our protocol. Please find below answers to all comments, with a reference to the manuscript edits. 

Thank you once more for considering our manuscript for publication in Plos One. 

Sincerely, 

Dr Julie Cayrol 

Reviewer 1

Reviewer Comments Author Responses

1. It is unclear if the proposed study will be an epidemiological cross-sectional survey, as mentioned in the title and background section, or it is a mixed method study consisting of cross-sectional survey and a qualitative study, as mentioned in the Design (line 265) and Follow-up (since line 364). Please make this clear, and make it consistent throughout the manuscript. 

Thank you for raising this. We have amended the document to reflect that this is a cross-sectional survey with some open-ended questions, as opposed to a true mixed-methods study, given the small number of qualitative data points.

2. The Abstract had sufficient background information, but lack basic information of the proposed study, such as aims, study design, participants and recruitment methods, data collection and analysis and outcome distribution. Please add.

Thank you for these suggestions. Changes have been made accordingly in the manuscript: aim, study design, recruitment methods, data collection, data analysis, outcome distribution have been added. Please kindly note that a description on study participants already existed in the abstract (line 47). 

3. Study outcomes (primary and secondary) were not presented in this protocol. Please add.

This has been added. Please see section under “Outcomes and dissemination”, page 22, line 810. 

4. Line 259-63 of page 13, the two secondary objectives were not technically objectives, but more like implications for future research. 

This has been reflected in a paragraph now named “Implications for future research”. (page 22, line 816). The paragraph on aims has been changed accordingly (page 13, line 506)

5. Line 409-415, contents regarding the sample size estimation need to be rigorously revised after consulting a statistician. It mentioned that “would provide a 95% confidence interval for the mean of a quantitative variable with a margin of error of ± 0.11 standard deviations”. Please note that quantitative variables with different variance require varied sample size to achieve a stable estimate of 95% CI (the more various, the bigger sample size need to be). Therefore, the authors need to specify, as well as to justify, their estimates on the means, SD, error level and anticipated response rate for the sample size estimation. For a global survey, I highly doubt the sample size of 2000 is justifiable. 

Apologies, a mistake was made within this sentence when we had updated some calculations. The subgroup sample size should have been 300. As a result, the sentence should be: “Within a particular subgroup, a sample size of 300 independent responses…” (page 21 line 778).

We agree that the sample size required to achieve a certain absolute confidence interval (CI) width would be larger for a more variable outcome. However, because we are collecting many outcomes, with differing and unknown levels of variability, our sample size justification considers the relative width of the resulting CIs, expressed on the scale of the standard deviation of the outcome. This acknowledges that outcomes with greater variability will have wider CIs, but they will achieve similar precision relative to the variability of that outcome in the sample.

In terms of the sample size, given this is the first global study of its nature, with broad primary objectives, we believe that any amount of data is justifiable and will allow us to learn about the experience of different populations living with cancer. 

6. Line 181-2 of page 9: “Limited data from LMIC has found that the financial impact of cancer is as significant as in HIC”. I doubt if this is the fact, as it’s widely known that people from LMICs generally have poorer access to healthcare services due to financial affordability of costs for cancer treatment. 

This was an error on our part, with the intention of saying, “at least” as significant as in HIC. This has now been changed to avoid confusion. (page 10, line 368)

7. Line 250-1 of page 12, the description of the study aim as “The aim of this study is to better understand and document the lived experience of people affected by cancer” is quite vague and lack of focus. 

The objectives of the study have been updated accordingly (page 13, line 506). 

8. Some expressions of the manuscript are ambiguous and need to be more precise, below are some examples:

1) Line 198-9 of page 10: “this loss is likely to be significantly higher today.” it’s good to give your reasons for the assumption, e.g., increased cancer prevalence and aging of the population?

2) line 206 of page 10: “Australian data suggests that for adolescent and young adults diagnosed with…”. And you also mentioned at line 211 of page 11:”… adolescents and young adults diagnosed with cancer total…”. Globally, there are no commonly accepted age ranges for “adolescents and young adults”, so it’s important for you to give the specific age ranges of participants in these two cited studies.

3) Line 220-1 of page 11: “…, and single mothers are at greatest risk (56–58).” This sentence is incomplete, at the greatest risk of what? financial hardship? Income loss?

4) What are the differences/similarities of financial toxicity/hardship?

5) Line256 of page 12:” died from cancer in a diversity of settings (HIC and LMIC).” By “settings”, do you mean in countries with different levels of development?

6) Line 265-7 of page 13: “This study is an observational, descriptive, mixed method cross-sectional study, distributed globally by the World Health Organization (WHO) Headquarters. No particular country or region is specifically targeted for inclusion.” Do you mean “This study is an … survey, which will be distributed among all member countries or regions of The World Health Organization (WHO), with no particular exclusion of any country or region.”

7) Line 314-5 “the questions are not targeting specific countries. For this reason, ethical approval from other countries was not sought.” A more rigorous description could be “survey questions are designed for general use across HICs and LMICs, so no ethical approval will be sought from individual participating countries.” 

Thank you for these very helpful suggestions. The manuscript has been modified throughout to reflect these points. Please see below, point by point: 

1) Page 11, line 412. Added “due to increased cancer prevalence”

2) Page 11 line 421. Ages clarified. 

3) Page 12, line 452. Specified “Risk of employment loss”.

4) Please note these terms are used interchangeably in the literature, This is now clarified in the manuscript with some references to support this point. Page 9, line 329.

5) Page 13, line 510 has been simplified to avoid confusion and this has also been changed throughout the manuscript. 

6) Yes, we are not distributing specifically to certain countries. This has been clarified. Page 13, Line 514. 

7) Thank you, see page 16, line 638.

9. The introduction section focused on the findings of previous studies on three major impacts of cancer on different groups of populations. Nevertheless, this section failed to adequately explain why the proposed study is needed.

It is necessary to systematically review existing literature on relevant topics, e.g. what existing surveys/studies have investigated, the methodologies they used, main findings (this was included in the current manuscript). Additionally, it is important to point out the gaps in current literature and how your proposed study will meet or help to narrow these gaps. It’s also crucial in this section to clarify the strengths of your proposed study compared to previous studies. Otherwise, it would be questionable in the necessity or significance of the proposed study. For example, if there is no adequate novelty of your proposed study, then to conduct a systematic review may be enough to get a comprehensive knowledge of this topic.

Thank you for your comment. While we have not performed a systematic review of the literature, we had thoroughly reviewed the literature on the three major impacts described in the literature and repeated this on the occasion of this review. We have included mainly literature review articles, including systematic reviews and meta-analyses. This is now more precisely addressed in the introduction, highlighting the lack of data from LMICs. We have also added some details on literature gaps, study justification, and strengths of the study. 

10. Line 277-8 of page 13: “finalising the survey, and interpreting the data after data collection.” This is confusing. Do you mean you have collected any survey data? 

Thank you for pointing this out. This sentence has been removed, and details now include what the Committee has completed so far. Page 14, Line 580

11. Authors misused “survey” as “survey questionnaire” in multiple places, e.g., line 299 of page 14 “The survey was developed by the Steering …”, line 304 “… the survey may have missed.” The “survey” in these sentences should be “survey questionnaire”. Please check and correct thoroughly. 

These changes have been made throughout the document, please see tracked changes. 

12. Line 307 “Table 1 summarizes the structure and content of the survey.” Here, it’s essential to report the main information of Table 1 in texts to facilitate understanding of readers. Also, it probably will be difficult to present all the information of Table 1 in the main text when it comes to publication. I suggest keeping the outline and main contents of Table 1 in the main text, while keeping the completed version as a supplement. The title of Table 1 contains repeated information, consider to revise it. 

Thank you for this very helpful suggestion. Now page 15, line 624. The title of table 1 has been simplified. A condensed version has been created to be included within the text, and the full table can be accessed as supplementary material. 

13. For such a global survey, it’s crucial to take into consideration of the priorities and special needs of people from health resource constrained countries, from non-English speaking countries, or with special cultural beliefs of cancer (e.g., indigenous people). This was touched in line 317-20 briefly, but it’s not clear how these important aspects will be addressed in the study design. Please add relevant information. Similarly, more information is needed how the study will be implemented in country with different languages, cultures and health systems? Will limited adaptation of the survey questions allowed? Will translation in local language available? 

Thank you for this important point. The survey was established amongst a group that included people from LMIC and was also shared with stakeholders working or originating from LMIC, to ensure content was appropriate to context and culturally sensitive. Additional details have been added in the manuscript about the process for translating the survey questionnaire. See page 17, line 684. The survey questions will also be reviewed by an expert in Health literacy within WHO to ensure the language is appropriate in terms of complexity for health resource constrained countries. However, adaptation for specific countries will be limited at this stage to allow for consistency during data analysis. The study will be disseminated through local stakeholders in different countries. 

14. Some people in under-developed countries may have no or very limited access to medical treatment. It can be important to know if they are able to receive treatment, if so, is this treatment adequate (i.e., to have the necessary treatments that they need); if not, what are the key barriers. 

Thank you for raising this important point. However, while tremendously important, it is not the focus of this study to assess the adequacy of the medical / supportive care treatment received and its barriers. This may perhaps be studied in future studies. 

15. Line 366-7 “… by providing their contact details (name, email address)”. Authors may consider including phone number as a contact method, as this is more commonly and frequently used than email. By including name and email only, it may be at a risk of omitting certain groups of people who don’t use/check email regularly, like those from low socioeconomic status. 

Thank you for this wonderful suggestion, indeed apps like ‘whatsapp’ are more commonly used in some resourced constrained settings than email. We have reflected this change (now page 18, line 708). 

16. Other minor changes

1) Line 268, to remove “(Figure 1)” from the subtitle, as you will refer to it in the main text.

2) All HIC and LMIC to be HICs and LMICs

3) At line 99-102 of page 6:” A systematic review of the prevalence of depression and anxiety among long term cancer survivors found that anxiety was more prevalent among cancer survivors compared to healthy controls, and was common among spouses of cancer survivors”. Here you may specific the definition of “long term cancer survivors”, as well as the prevalence of the two study groups.

4) Line 423 “Additional limitations to a cross-sectional design include difficulty deriving causal relationships between a cancer diagnosis and participants’ current or past lived experience”. If this is going to describe the current situation, rather than to perform causal inference, then the mentioned limitation would not be an issue.

1) Change made (line 515 title Governance)

2) This change has been made throughout the manuscript

3) This information has been added (page 6, line 125)

4) We agree with the reviewer, and have reworded this sentence. The study indeed focuses on describing and will not attempt to derive causal relationships between a cancer diagnosis and participants’ current or past lived experience. (page 21, Line 796)

Reviewer 2

Reviewer Comments Author Responses

1. The authors are highlighting a lack of available global data on the topic in general and in LMIC especially. The topic clearly is of great importance and the study will create helpful and interesting information for institutions and individuals working with cancer patients or in health system planning.

Thank you for your comments. 

2. The background section starting as line 62 gives a comprehensive overview on the psychosocial and financial issues cancer patients and relatives are facing according to the literature. While the structure is clear and good to follow, it is quite lengthy by writing style and the information could be conveyed in a more compact manner.

We appreciate this comment. We have tried to condense the introduction and shorten the introduction for readers’ ease. 

3. The highlighting of the lack of data from LMIC both in abstract (lines 40; 50) and main text (e.g., lines 226; 240) and the studies aim to add new information to the literature are positive. The inclusion of high-income countries and LMIC will help determining differences between countries and regions. Nevertheless, the representation of the LMICs seems it will be lower in the study. Since the authors stated the special attention needed in LMIC, further inclusion of such settings in the ongoing planning process seems advisable if resources exist. It is advisable to follow the stated further inclusion of stakeholders from underrepresented regions (lines 280 - 286). While there is an international collaboration working on the project, affiliate members from regions Asia and South- and Central America seem missing, especially since the questionnaire will be available in Spanish and Chinese. 

Thank you for raising this point. Indeed, representation from LMIC was lacking in the pilot study. This time, we have been working on a better dissemination strategy to target the various regions of the world (see S2 table). We are currently conducting a round of stakeholder feedback from members of various global cancer organisations with representation in the mentioned regions (South East Asia, Central and South America). These stakeholders have also agreed to help with participant recruitment/ study dissemination. 

4. Lines 281 to 284 (invite additional members as needed, including, for example, underrepresented groups of people with lived experience (e.g., fathers, culturally diverse people, First Nations people): It is a challenging but important endeavor. I suggest making and publishing a plan how to deal with this situation in a structured manner. This will also increase transparency. 

The first version of the study had a large representation from HIC and women. We have been reaching out to individuals amongst different organisations and institutions to increase study dissemination. This has been added in the ‘Timeline’ section, page 19, line 725. Please see list of stakeholders in S2 table. 

5. Moreover, the recruitment plan (lines 353 ff.) seems to involve effective strategies with WHO support and via other institution. It might be helpful to publicize a strategy on how to recruit participants from the worlds region good receive adequate representation of the regions. Otherwise be more specific about the target regions.

Thank you for this suggestion. We have added a list of stakeholders from diverse regions who will help with study dissemination and participant reach, to ensure adequate representation from different regions. (Page 19, line 732)

6. It would be interesting to learn more about the methods of translating the questions into the different languages. 

Thank you for making this important point. We have added some details on how the translation process will be undertaken. (page 17, line 684)

7. Data collection seems to be anonymously if the participant does not agree for a follow-up. How is personal data handled for those cases? 

Indeed, data collection is anonymous, no personal details are collected through the survey and there is no way to link survey results to participants. Interested participants can agree to send their details to be contacted for follow-up. Personal data will be linked to a newly created personal ID. This confidential document will be stored on WHO servers with access only by the study team. This data will be destroyed after 5 years. (please see page 20, line 745)

8. Learning more from the participants in qualitative follow-ups seems very informative. It would be interesting to learn in more detail about the methods planed for qualitative follow up interviews especially since the number of participants could become very high.

Thank you for your comment. While we agree follow-up interviews would be extremely valuable, we have not yet finalised any plans for specific follow-up studies. Therefore we have chosen not to include any additional details, as these plans may change. Thank you for your understanding. 

9. The planned sample size is clearly stated (n=2000) and calculates for a subgroup of 1000 participants indicating a binary subgroup analysis (lines 409 ff.). However, the planned target groups are 4 (lines 324 ff.) and multiple further subgroup analyses are mentioned (disease and age groups, countries: lines 398 ff.; time since diagnosis line 426). It would be interesting to learn more about the calculations giving these circumstances as well as the planned definitions of the disease/age subgroups (there are multiple cancer entities, that largely differ in the way they affect patients’ health and with it their individual needs). This would support the mentioned development of the more detailed analysis plan (line 428-430).

Apologies, as mentioned above, a mistake was made when we had updated some calculations. The subgroup sample size should have been 300. As a result, the sentence should be: “Within a particular subgroup, a sample size of 300 independent responses…” (page 21 line 779). 

We appreciate the reviewer’s comment, and have some preliminary subgroup definitions (adult vs childhood cancer; types of cancer will be grouped in solid tumours vs leukaemias vs high-grade brain tumours and low-grade brain tumours). However we have not yet finalised our definitions of subgroups and plan to do so before performing our data analysis. We would therefore prefer not to include definitions in the protocol before these are finalised. 

10. Concerning the distribution, those seem reasonable. It might be fair to participants and seemingly manageable to offer the option of sending the planned easy to understand summary of the studies’ findings at the end of the online questionnaire. 

Thank you for raising an excellent point which was mentioned by some stakeholders and participants themselves. We will add this as an option and will look forward to informing participants who wish to receive an easy to understand summary. 

11. Overall, the scientific community can look forward to see results from this upcoming study.

Thank you. We look forward to completing this study and sharing the results with our global scientific community.

---

## [Editor Report · Decision Letter 1]

2 Nov 2023

The lived experience of people affected by cancer: A global cross-sectional survey protocol

PONE-D-23-17936R1

Dear Dr. Cayrol,

We’re pleased to inform you that your manuscript has been judged scientifically suitable for publication and will be formally accepted for publication once it meets all outstanding technical requirements.

Kind regards,

Sefonias Getachew, MPH, PhD

Academic Editor

PLOS ONE

---

## [Editor Report · Acceptance letter]

8 Nov 2023

PONE-D-23-17936R1 

The lived experience of people affected by cancer: A global cross-sectional survey protocol 

Dear Dr. Cayrol:

I'm pleased to inform you that your manuscript has been deemed suitable for publication in PLOS ONE. Congratulations! Your manuscript is now with our production department. 

Kind regards, 

on behalf of

Dr Sefonias Getachew 

Academic Editor

PLOS ONE